# Three-Dimensional Convolutional Neural Network for Ultrasound Surface Echo Detection

**DOI:** 10.3390/s25165033

**Published:** 2025-08-13

**Authors:** Mario Muñoz, Adrián Rubio, Marcelo Larrea, Jorge F. Cruza, Jorge Camacho, Guillermo Cosarinsky

**Affiliations:** 1Institute for Physical and Information Technologies, Spanish National Research Council, 28006 Madrid, Spain; adrian.rubio@csic.es (A.R.); jorge.f.cruza@csic.es (J.F.C.); j.camacho@csic.es (J.C.); 2Electronic Department, Universidad de Alcalá, 28805 Alcalá de Henares, Spain

**Keywords:** time of flight, surface detection, non-destructive testing, convolutional neural network, deep learning

## Abstract

Ultrasound array imaging frequently employs a coupling medium to facilitate wave transmission from the transducer to the target component. Surface echoes, identified by their high-amplitude peaks, are crucial for determining the Time of Flight (TOF) in each channel, which is essential for deriving imaging focal laws. Accurate TOF measurement is vital in numerous applications, such as Non-Destructive Testing (NDT) and medical imaging. Conventional methods, such as threshold crossing and peak search, are highly sensitive to noise and spurious signals, therefore, more robust estimation techniques are needed. This study explores the application of a deep 3D Convolutional Neural Network (CNN) to detect surface echoes in Full Matrix Capture (FMC) ultrasound data. The CNN was trained on signals obtained with a matrix array and a set of reference components, utilizing a robotic arm setup to ensure precise probe positioning. Theoretical TOFs were computed based on the setup geometry to generate labeled training data. Test results indicated that the CNN model, which we have called DeepEcho3D, closely aligned with the ground truth and significantly reduced TOF estimation outliers (up to 98%) compared to traditional methods, demonstrating its potential for improved accuracy in surface echo detection.

## 1. Introduction

Ultrasonic Testing (UT) is a prevalent technique in the field of Non-Destructive Testing (NDT). In particular, the use of array type probes [1] is a very powerful tool for internal imaging of industrial and structural components. Array probes enable many imaging modes, like the classical Phased Array (PA), and more advanced methods like Total Focusing Method (TFM) [2] and Plane Wave Imaging (PWI) [3,4,5].

UT requires proper coupling between the transducer and the component under test. This is often achieved through immersion testing, where both the component and the transducer array are submerged in water. This method is particularly useful for components with curved surfaces and/or varying geometries during scanning, as it can adapt to these geometrical changes.

However, immersion testing poses challenges in computing imaging focal laws due to ray refraction at the component surface. For each array element and each image point, the rays between them must be computed. This requires a detailed understanding of the system geometry, including the surface shape and the Probe Location and Orientation (PLO) relative to the surface, in order to use either Fermat’s principle or Snell’s law to compute the ray incidence point. This challenge leads to the development of adaptive imaging algorithms [6,7], also known as autofocusing [8,9,10], where the ultrasonic signals are used to estimate the geometry before generating the image.

These algorithms can be categorized into two types. The first type utilizes Total Focusing Method (TFM) imaging in water to detect the surface as a high-intensity region [11,12]. The second type relies on measuring the Time of Flight (TOF) of surface echoes from individual array elements to compute surface points [8,12,13,14,15] or construct a geometrical model [16].

For example, McKee et al. [11] use TFM to image doubly curved surfaces with a matrix array. A search procedure is then applied to identify the pixels with locally maximum amplitude in the 3D image, which correspond to surface points. Similarly, Robert et al. [12] present various surface estimation methods for linear arrays, including a TFM-based approach similar to [11] and TOF-based methods. Matuda et al. [14] also introduce two different TOF-based methods along with a hybrid approach that first detects the surface echo and then applies an imaging algorithm along a circular arc, rather than computing a full image as in TFM-based methods. Like the methods in [12], these approaches are specifically designed for linear arrays and 2D imaging and require calculating an image or sub-image, only for obtaining the surface shape.

The TOF-based methods described in [8,12,13,14,15] are local approaches, where the TOF from an array element and its nearest neighbors are used to estimate a single surface point. In contrast, a global TOF-based method is introduced in [16], where TOFs from all array elements are fitted to a parametric model of the system geometry. All TOF-based methods require detecting the surface echo to measure its TOF. Therefore, for this second category of algorithms, a robust TOF measurement technique is essential.

Surface echoes ideally appear in the ultrasound signals as the first arriving pulse, and the simplest method for its detection is threshold crossing. However, other pulses might reach the transducer earlier, such as those coming from bubbles in water or surface waves propagating from the emitter element in an array (Figure 1). In the case of a concave surface, side lobes or grating lobes can also create reflections that arrive earlier than the main lobe echo, which is the target of the detection method. These spurious echoes, or noise, might cross the threshold, generating an erroneous measurement.

Besides threshold crossing, there are many methods for pulse localization and TOF measurement [17,18,19,20]. In [17], three methods are developed for range measurement applications. These approaches compare the received signal to a reference signal, which can be either a theoretical pulse model or an echo measured from a calibrated reference. In [18], various methods are evaluated for Acoustic Emission (AE) signals, including a modified cross-correlation technique and two approaches based on the Continuous Wavelet Transform (CWT). In [19], a set of seismic traces is processed as an image, and a connectivity-based algorithm is employed to identify echoes. In [20], the challenge of detecting a reflected echo that overlaps with the transmitted signal is addressed using a Time–Frequency–Amplitude (TFA) distribution and a signal decomposition technique known as the Synchro-Squeezed Transform (SST).

All of these methods suffer from the aforementioned difficulties. Therefore, developing better methods to address these challenges is crucial, as TOF measurement is used not only for UT but also for many applications involving acoustic signals, such as Acoustic Emission (AE) testing [18], seismic traces [19], speech analysis [21] and ultrasound tomography for medical imaging [22]

Machine Learning (ML) and specifically Deep Learning (DL) are promising as tools for many signal and image processing applications. Research on ML and DL methods for Non-Destructive Testing (NDT) is a rapidly expanding field [23,24,25]. In [23], a review is presented of the current state of research on DL applications in NDT. For instance, in [24] a 2D CNN is used to classify ultrasound images of welded components as either defective or not defective. In another example [25], three methodologies were developed by combining Laser Doppler Vibrometer (LDV) measurements with ML approaches for the testing of Carbon-Fiber-Reinforced Polymer (CFRP) plates.

Recent research further highlights the trend of combining advanced imaging with ML. For instance, Miorelli et al. [26] use deep learning on multimodal TFM images for crack characterization, leveraging data augmentation from physics-based models. Similarly, Tu et al. [27] employ advanced signal decomposition techniques for improving TFM imaging of surface-breaking cracks. While these powerful methods focus on interpreting the final TFM image to characterize defects, our work addresses the preceding, fundamental challenge of robustly detecting the component’s surface. Accurate surface detection is a critical prerequisite for generating the high-quality TFM images that these characterization techniques rely on, particularly in adaptive autofocusing scenarios.

TOF measurement, the particular problem we deal with in this work, is addressed in [28] for AE signals using an improved Akaike Information Criterion (AIC) based on a 1D Convolutional Neural Network (CNN), yielding excellent accuracy and stability. A 1D CNN is also applied in [29] for microseismic monitoring. Another geophysics example is found in [30], where a U-net [31] 2D CNN is applied for image segmentation and a Recursive Neural Network (RNN) for first arrival picking. The authors treat a set of seismic traces from a sensor array as an image and segment these into two categories: background (before first echo) and signal (after first echo).

In the present work we also treat the problem as a segmentation task, but we use volumetric data from a matrix probe instead of 2D images. While we base our network architecture on the V-net [32], a 3D extension of the U-net, we go beyond simply adopting a standard architecture. We systematically optimize the network’s hyperparameters using the Hyperband algorithm [33], creating a tailored architecture specifically adapted to the challenges of surface echo detection in 3D ultrasonic data.

One of the biggest challenges for the successful use of DL is the need for a large annotated dataset for model training. In the cited works [22,26,27,28,29,30], the labeling is performed manually, i.e., the first echo is manually picked by a human expert. To overcome this limitation, we introduce a novel automated labeling method, which, to our knowledge, has not been previously applied to this type of data. Our approach leverages a physical model to compute the surface echo TOF based on known geometry parameters (PLO and surface shape) under controlled laboratory conditions. This automation not only eliminates human bias and variability but also significantly accelerates dataset generation.

## 2. Methods and Materials

Figure 2 presents a flowchart of the methodology followed, which will be explained in detail below.

### 2.1. Data Preparation

#### 2.1.1. Experiment Setup

A 6-axis collaborative robot (Universal Robots, Odense, Denmark) equipped with a 3 MHz, 1 mm pitch, 11 × 11 matrix array ultrasound probe (Imasonic, Voray-sur-l’Ognon, France) attached to its arm was used for the experiments. The ultrasound instrument used was a 128-channel full parallel system (DASEL S.L., Arganda del Rey, Spain). This setup allowed precise control over the positioning and orientation of the probe relative to the test components. The components used in the experiments included a flat aluminum plate, a steel sphere with a 9.5 mm radius, and four cylinders, CYL1, CYL2, CYL3, and CYL4 (Figure 3a). CYL1 and CYL3 are convex aluminum cylinders, while CYL2 and CYL4 are aluminum blocks featuring a concave cylindrical surface. In Table 1 the use of each component in the study is shown. These specific components were chosen as they represent the fundamental canonical shapes (flat, convex, concave), allowing for a systematic evaluation of the model’s ability to learn basic geometric features. The hypothesis is that the CNN could then generalize to a wide variety of component shapes.

During the experiments, each component was placed on a flat surface, which served as the reference plane for measuring the Probe Location and Orientation (PLO) following the method explained in our previous work [16]. In the case of the aluminum plate, the component itself acted as the reference plane.

#### 2.1.2. Data Acquisition and Preprocessing

Accurate referencing during acquisitions requires defining a coordinate system, known as the World Coordinate System (WCS), to track the probe’s location and orientation (PLO) relative to the component surface. This reference is established following the methodology of [16], using Cartesian axes (*x*, *y*, *z*) and rotations parametrized by three angles *θ* = (*θx*, *θy*, *θz*). The components were tested in an immersion setup from various PLOs, resulting in multiple FMC datasets per test piece. Before using the data as input for the model, a 62-coefficient filter with a [0.5, 6] MHz pass-band was used to eliminate out-of-band noise and signal offset.

Each Full Matrix Capture (FMC) is composed of 121 sub-images corresponding to each emitting element. The signals were acquired with a sampling frequency of 40 MHz and have 1000 samples, which is enough range to contain the surface echo for the used PLOs. This sampling frequency was set above the Nyquist limit (assuming a 100% bandwidth centered at 3 MHz), ensuring an accurate digital representation of the signal’s waveform. Thus, for each emission an 11 × 11 × 1000-voxel volumetric image is obtained. This data volume will be the input to our CNN. For the representation of the images, the three-dimensional matrix is reshaped into 121 × 1000 pixels in order to paint a two-dimensional image as shown in Figure 1.

#### 2.1.3. Ground Truth

A huge amount of annotated data is needed to train a CNN for our purpose. In this case, a method of automatic labeling was implemented to achieve that goal.

As the shape of the component being tested and PLO relative to the component are known, it is possible to compute the theoretical Time of Flight (TOF) for each pair of array elements. This calculation involves identifying a point *G* on the surface *S* (Figure 4) that satisfies the law of reflection. For complex surfaces, this process typically requires numerical root-finding techniques. However, for certain simple geometries, approximate closed-form solutions can be used. In our previous work [16], we developed an approximation method specifically for cylindrical and spherical surfaces. This method is utilized in this study to determine the theoretical TOF for each signal within the FMC. With this calculation, it is possible to determine the theoretical sample number ${Idx}_{\mathrm{true}}$ of the surface echo arrival time. The Ground Truth (GT) value is generated by:(1)$$GT=\left\{\begin{array}{c} 0,\;\;ix<{Idx}_{\mathrm{true}}\\ 1,\;\;ix>{Idx}_{\mathrm{true}} \end{array}\right.$$

In this context, the index ix represents the sample number. Any sample with an index less than ${Idx}_{\mathrm{true}}$ is labeled as 0, indicating that it is above the surface echo (out of the component). On the other hand, any sample with an index greater than or equal to ${Idx}_{\mathrm{true}}$ is labeled as 1, which means that is below the surface (inside the sample). This method ensures that our model accurately identifies surface echoes based on the theoretical TOF, helping to discriminate against noise and spurious echoes generated by lateral lobes, bubbles in water, or probe surface waves, as shown in Figure 5.

### 2.2. Base CNN Architecture Selection

In this study, we implemented a 3D Convolutional Neural Network (CNN) to detect surface echoes in Full Matrix Capture (FMC) ultrasound data. The decision to use 3D CNNs was driven by their proven effectiveness in capturing spatial and temporal hierarchies and extracting relevant features from complex multidimensional data. Given the 3D nature of ultrasound signals captured by a matrix array, a 3D CNN is naturally suited to process this type of data. Its 3D convolutional kernels can directly learn spatial relationships in all three dimensions. This is an important point, as the surface echo will typically exhibit a consistent pattern across neighboring samples, corresponding to the surface shape. A 1D CNN applied to individual A-scans would miss this crucial context, and a 2D CNN might be appropriate for a linear array but will not capture the echoes’ relations on both principal directions of a matrix array.

Based on [32], the V-net architecture, a 3D extension of U-net, was chosen. Its encoder–decoder structure, with skip connections, allows for capturing both local details and global context, essential for our task. This architecture was specifically tuned to handle the 3D nature of FMC data, making it suitable for detecting surface echoes in ultrasound signals.

### 2.3. Parameter Optimization

In the context of deep learning, hyperparameters (such as number of layers, size of each layer, learning rate, etc.) are critical as they define the architecture of the neural network and the training process. For our model, we have identified several key hyperparameters that influence the model performance. These hyperparameters include the size of the convolutional kernels, the number of convolutional layers (or blocks), the pooling size, the learning rate, and the type of loss function used.

The hyperparameter tuning process consists of a search in the hyperparameter space with the objective to optimize the model performance. For each selected point in the hyperparameter space, the corresponding model is trained and tested with some chosen metrics. There exist many algorithms for the search process. In the case of a small hyperparameter space, an exhaustive search can be performed. In our case this would be excessively time consuming, thus we chose the Hyperband search algorithm [33].

#### 2.3.1. Hyperparameter Definition

Number of Convolutional Blocks: This impacts the depth of feature extraction. It was tuned between 2, 3, 4, or 5 convolutional blocks to ensure sufficient depth for feature extraction without overfitting. Each block is composed of 5 layers: 3D convolution, spatial dropout, batch normalization, 3D convolution, and another batch normalization layer.Convolutional Filter Size (Kernel Size): This determines the receptive field of the convolutional layers and affects how the network captures spatial features. It was adjusted to capture the most relevant features along the time axis with values between [3, 3, 6] and [3, 3, 21].Number of Filters: The number of filters in each convolutional layer influences the capacity of the network to learn diverse features. It was optimized with values typically ranging from 8 to 16 filters per layer with the aim of not oversizing the network.Pool Size: This influences the reduction of spatial dimensions over the CNN, and it is optimized to find the correct dimension reduction to be applied in MaxPooling layers. The x- and y-axes were tuned with values 1 or 2, and the temporal axis was tuned with values 2, 4, or 8.Loss Function: The loss function plays a significant role in guiding the network’s learning process towards minimizing prediction errors. We experimented with three different loss functions to find the most effective one for our specific application: binary cross-entropy [34], Tversky [35], and Dice [36] losses, which are typically used in segmentation problems [37].Learning Rate: This was chosen to ensure stable and efficient training, balancing the speed of convergence with the stability of the learning process.

#### 2.3.2. Hyperparameter Search

To optimize the hyperparameters, we employed a systematic hyperparameter search using the Hyperband algorithm [33]. It is an algorithm that leverages the principle of adaptive resource allocation and early stopping to efficiently explore a wide range of parameter configurations and allows for speeding up the search. The hyperparameter space defined in the previous section has more than 1000 configurations. Hyperband dynamically allocates more resources to the most promising configurations.

With the defined parameters, candidates are obtained. Each Hyperband iteration selects a random subset (30 configurations for the computing resource budget used in our case) and performs a sequence of training rounds where the worst models are discarded. Six hyperband iterations were used, with the aim of minimizing the *Idx Error* metric of the validation dataset.

The *Idx Error* is defined as follows:(2)$$Idx\;Error=\max(\left|Idx_{pred}-Idx_{true}|\right)$$
where $idx_{pred}$ represents the predicted surface echo indexes with an output threshold of 0.5 and $idx_{true}$ represents the ground truth surface echo indexes.

### 2.4. CNN Final Selection

After identifying and optimizing the hyperparameters using the Hyperband algorithm, the next step was to select the most effective CNN architecture from candidate models with the following approach:

#### 2.4.1. Initial Preselection

From the initial set of configurations, after applying the Hyperband search algorithm, the top 5 best-performing models were preselected based on their validation *Idx Error* metrics. This preselection aimed to narrow down the candidate pool to a manageable number of models for further in-depth analysis, given the similar metric results obtained with these options.

#### 2.4.2. Retraining and Stability Analysis

To ensure the robustness and consistency of the selected models, we retrained the top five models identified in the preselection phase five times each. This step was essential to assess the variability in model performance due to the stochastic nature of the training process. By evaluating the models across multiple runs, we aimed to identify architectures that consistently demonstrated low index errors and a reduced number of outliers.

During the retraining process, several performance metrics were recorded, including:

*Idx Error*: As previously defined, the *Idx Error* measures the maximum absolute difference between the predicted and ground truth surface echo indexes. A lower *Idx Error* indicates more accurate surface echo detection.

Number of Outliers: We quantified the number of significant deviations (outliers) in the predicted TOF compared to the ground truth. This metric is crucial for assessing the reliability of the model in practical applications. We categorized outliers into two types:Outliers where the surface echo is not detected because the network threshold (0.5) is never crossed for that particular image line (type 1). In the case of the standard threshold crossing method this type of outlier corresponds to the signal never crossing the predefined threshold level.Outliers where the index error exceeds a predefined number (max_idx_error) of samples (type 2).

Monitoring these metrics during the retraining phases helped to ensure that the model was learning effectively and not overfitting to the training data.

#### 2.4.3. Final Model Decision

Based on the retraining and performance analysis, we selected the CNN architecture that consistently exhibited the best performance across all metrics.

By following this comprehensive CNN selection method, we ensured that the final model was not only the best performing in terms of accuracy but also robust and reliable for detecting surface echoes in Full Matrix Capture (FMC) ultrasound data. The final architecture resulting from this process, which will be used for the remainder of our study, is detailed in the Section 3.

### 2.5. Training

A dataset of 6000 annotated ultrasound volumes was used to train our 3D CNN divided into training, validation, and testing sets with a 70:15:15 ratio. Training was carried out using 2 NVIDIA 2080 Ti GPUs in parallel. The model was trained for a maximum of 70 epochs, with a batch size of 64 volumes. To prevent overfitting and select the best-performing model, we employed an early stopping strategy. A callback function monitored the validation loss (*Idx Error*) after each epoch. Training was automatically stopped if the validation loss did not improve for 6 consecutive epochs (patience = 6). Furthermore, we saved the model weights at each epoch where the validation loss improved. Figure 6 shows the learning curves for both the training and validation sets, plotting the *Idx Error* as a function of the training epoch. This early stopping strategy ensures we choose a model that generalizes well, rather than one that is overly specialized to the training data. In addition to the *Idx Error*, the number of outliers was also tracked during training as a secondary performance metric to assess the model’s ability to handle noisy and distorted signals.

### 2.6. Test

For the test process, we used acquisitions from components not included in the training dataset to evaluate the performance of the model in terms of index error and the number of outliers. The new test pieces included a concave metallic cylinder (CYL4) with a 40 mm diameter, two convex metallic cylinders with a 35 mm diameter (CYL2) and 12 mm diameter (CYL1), respectively.

By testing on these new pieces, we aimed to assess the model’s generalization capability and robustness. The evaluation focused on ensuring that the model maintained a low index error and effectively minimized the number of outliers, demonstrating reliable performance on unseen data. In the Section 3, all metrics and graphs are shown in detail.

## 3. Results and Discussion

### 3.1. Parameter Optimization Results

After executing six iterations of Hyperband hyperparameter optimization tuner algorithm, a total of 237 valid candidates were monitored, focusing on the validation index error and the total number of outliers, taking into account the two different types mentioned in the previous section. In Figure 7, the result obtained for each trial is shown, where the *x*-axis represents the validation error (val_idx_error) and the *y*-axis shows the sum of the two types of outliers for the test dataset (val_n_out_1 + val_n_out_2).

We observe a clear trend in the trial results. As the validation index error increases, the number of outliers also increases. This behavior is expected, as a model with lower validation error should be more accurate in detecting surface echoes and thus generate fewer outliers.

However, there are some trials that exhibit high validation index error and a significantly larger number of outliers. These cases indicate hyperparameter configurations that are not suitable for the model, resulting in lower accuracy and higher incidence of errors.

### 3.2. CNN Selection Results

It is worth noting that several configurations exhibit very similar behavior (Figure 7 right), which suggests that any of these configurations could serve as a viable solution. This redundancy in effective configurations provides some degree of flexibility in selecting the optimal model. To ensure stability and robustness, we selected five of these promising configurations for further training and evaluation, aiming to determine which one demonstrates the most consistent and reliable performance. To clarify the architectural differences between these candidates, their specific hyperparameter configurations are detailed in Table 2.

Figure 8 displays the results of the five best models after being trained five different times. Each model is represented by a different marker in the scatter plot, where the *x*-axis indicates the validation index error (val_idx_error) and the *y*-axis represents the sum of the two types of outliers (val_n_out_1 + val_n_out_2).

Each training run introduces randomness, which leads to variations in model performance even when the architecture remains the same. This inherent randomness affects both the validation index error and the number of outliers, as seen in Figure 8.

The val_idx_error of the models in Figure 8 is around five samples. As the sampling frequency was 40 MHz, a five-sample error corresponds to 125 ns. With a 1.48 µm/s propagation speed in water, the distance error is about 0.2 mm. The total number of outliers is around 0.06, that represents the average number of outliers per batch at the end of the training.

### 3.3. Final Decision

As outlined in our methodology (Section 2.4), the goal of retraining was to assess performance stability against the randomness of initialization, a key indicator of a robust model. To provide a quantitative basis for this assessment beyond the visual inspection of Figure 8, a stability analysis was performed. In Figure 8 each model is represented by a cluster of points. The centroid (mean [val_idx_error], mean [total_outliers]) of each cluster and the average Euclidean distance to this centroid for each model across its five runs were calculated. This average distance represents the performance dispersion, where lower values indicate higher stability. The results are summarized in Table 3.

The data in Table 3 quantitatively confirms the visual assessment. Model_3 exhibits the lowest average distance (0.09), which quantitatively proves it has the highest performance consistency and is, therefore, the most stable architecture.

Given its superior stability, Model_3 was selected as the final architecture. This configuration, now named DeepEcho3D and detailed in Table 2, represents the most robust and reliable solution identified through our evaluation process. Figure 9 illustrates the final architecture of DeepEcho3D.

The training process for the final DeepEcho3D model on the full dataset took approximately 3 h on two NVIDIA 2080 Ti GPUs (NVIDIA Corporation, California, United States). The inference time for a single volume is approximately 18 ms, demonstrating its suitability for practical analysis.

### 3.4. Test Results

The test procedure of DeepEcho3D on the new test pieces, which were not included in the training set, yielded the metrics of Table 4.

These high metrics indicate that DeepEcho3D performs exceptionally well in identifying and segmenting the surface echoes. The Dice coefficient and Intersection over Union (IOU) mean values are very close to 1, showing excellent overlap between the predicted and actual surface echoes. The high recall and precision mean values also suggest that DeepEcho3D is both sensitive (able to detect most actual surface echoes) and precise (with few false positives).

However, while the high Dice mean and IOU values are indicative of good performance, they might not be the best metrics for evaluating DeepEcho3D’s behavior, given the nature of our labeling approach where it is segmented from the surface echo downwards. For instance, if the model’s prediction closely matches the corresponding ground truth but includes a few pixels with a value of 1 in the area just before the surface echo, the number of misclassified pixels would constitute only a small percentage of the total. A high overlap results in a high concordance in classic segmentation metrics like Dice and IOU. However, a few outliers might be very detrimental for a surface reconstruction algorithm. Therefore, we believe that evaluating and comparing the model based on the number of outliers and the index error provide a more accurate representation of how well our solution aligns with the desired outcomes.

Figure 10a shows box plots for the index error across different solutions: threshold crossing with four different threshold values and DeepEcho3D. These threshold values were selected based on the noise background observed in the signals and were designed to span a range of practical fixed-level settings, from a low value (thr_50) that is more sensitive but susceptible to noise to a high value (thr_200) that is more robust but may miss weaker echoes. DeepEcho3D consistently exhibits a narrow range of errors, indicating its reliability and stability compared to the thresholding methods, which show wider variability and higher median errors. DeepEcho3D has a lower standard deviation, implying more consistent performance, and a smaller interquartile range, showing that the majority of errors are tightly clustered around the median. Table 5 presents the index error range containing 95% of the detected points, along with the percentage of outliers relative to the total number of A-scans and the standard deviation (std) of the index error for each solution.

Additionally, Figure 10b illustrates the number of type 1 and type 2 outliers for each solution. Type 2 outliers occur when the index error exceeds 40 samples (max_idx_error), and type 1 outliers represent instances where no surface detection occurs. The plot highlights that traditional thresholding generates a significant number of outliers. At higher threshold levels, it becomes more likely that a signal never crosses the threshold, leading to an increase in type 1 outliers. On the other hand, lower threshold values are more frequently crossed by spurious echoes that precede the surface echo, resulting in more type 2 outliers. A threshold set at 50% achieves a balance between the two outlier types, but the total count still exceeds seventy thousand, in contrast to the one thousand outliers produced by DeepEcho3D. Figure 10c compares the distribution of the number of errors and total outliers for various threshold values against DeepEcho3D. The histograms indicate that the traditional thresholding methods produce a significant number of outliers, while DeepEcho3D generates 1381, which represent 1% of the A-scans in the test dataset, as shown in Table 5.

Figure 11 shows an example of the behavior of DeepEcho3D compared to typical threshold solutions.

Figure 11a shows a B-scan image using the radio frequency signals of all the 121 channels. Figure 11b shows the corresponding DeepEcho3D output, where a clear segmentation similar to the ground truth mask is observed. In the detail presented in Figure 11c TOF detection of the threshold method and the CNN can be observed alongside the ground truth curve. While the detection of the surface echo by DeepEcho3D fits quite well with respect to the ground truth, the threshold method presents some outliers of both type 1 and type 2.

Figure 12 shows another example of the DeepEcho3D response for an especially difficult case where the surface echoes have a very low amplitude and the threshold method fails completely. On the other hand, DeepEcho3D shows quite good performance despite presenting some deviations. This case shows the noteworthy capability of the CNN model to catch very low-amplitude surface echoes and discriminate them from other neighboring echoes.

It is important to highlight that both the training and testing of our deep learning model were conducted using components with relatively simple surface morphologies. This choice was made to ensure a clear and controlled evaluation of the model’s fundamental capabilities in detecting and segmenting surface echoes.

However, a crucial direction for future research involves expanding the study to include components with more complex surface geometries, as well as oxidation, roughness, or slight corrosion. Investigating how the trained model performs on intricate and irregular surfaces will provide deeper insights into its robustness and adaptability. Such an evaluation will help determine whether training on simple geometries is sufficient or if it imposes significant limitations on the model’s generalizability. Furthermore, a future comparative study against other state-of-the-art deep learning architectures, such as Vision Transformers [38], would provide valuable insights into the relative strengths and weaknesses of DeepEcho3D.

### 3.5. TFM Autofocusing with the Detected Surface Echoes

The final goal of the surface echoes’ TOF measurements is to use them to estimate the system geometry for generating an image of the interior of the component. In this section we present two examples of TFM volumetric images.

On the one hand, we evaluate our approach with a TFM image of CYL1, which has three Flat Bottom Holes (FBHs) as shown in Figure 13. The geometry is estimated using the pulse-echo method described in [16], in which the geometrical parameters are obtained from a least squares fitting of surface echo TOFs given by the CNN.

Figure 14a,b display the pulse-echo B-scan image alongside the echoes (red dots) identified via threshold crossing. The threshold was determined as a multiple of the Root Mean Square (RMS) noise within the water section of the A-scans. Specifically, Figure 14a employs a threshold of 3 × RMS, while Figure 14b utilizes a threshold of 6 × RMS. Figure 14c shows the echoes detected by the CNN trained in the previous section, using the 0.5 standard threshold for the network output. It is clearly observed from these three figures how the CNN is much more robust to outliers than the simple threshold method. These figures also display the fitting curves (blue lines), demonstrating that the lower threshold case yields an unsatisfactory result.

Figure 14d,e show a D-scan view of the TFM volumetric images for each of the 6 × RMS and CNN cases. In the case of 3 × RMS, the geometry estimated by the fitting algorithm is excessively deviated from the real one and the image cannot be formed, showing a case of total failure of the threshold method. In the case of 6 × RMS, the image is significantly distorted in relation to the case where the CNN is used, in which the echoes are best detected (Figure 14c). The use of the 6 × RMS threshold induces a relatively high error in the geometry estimation, leading to the image distortions: FBH indications are displaced from their actual positions, and their shapes and amplitude change.

On the other hand, DeepEcho3D was evaluated on an iron component (Figure 15a) with three Side-Drilled Holes (SDHs), using the same probe as in the previous example. The component’s surface exhibited oxidation and roughness (Figure 15b), characteristics absent in the training data. Furthermore, the geometry of the inspected zone includes planar and cylindrical regions, a combination that is not present in the training data either. Figure 16a displays the pulse-echo B-scan image, overlaid with the echoes detected by DeepEcho3D (red dots). These echoes were used to reconstruct the surface shape and generate a TFM image (Figure 16b), revealing two of the SDHs and a portion of the component’s back wall. This demonstrates DeepEcho3D’s ability to generalize to a more realistic, unseen component. However, further work is required to thoroughly assess the model’s generalization capabilities on more complex components.

## 4. Conclusions

In this study, we developed and validated a deep-learning-based method for detecting surface echoes in ultrasonic signals used in Non-Destructive Testing (NDT) applications. By employing a customized 3D CNN architecture, we approached this problem as a volumetric segmentation task. Starting from the V-net architecture introduced in [32] for 3D medical image segmentation, we applied modifications to adapt it to 3D NDT ultrasound data and fine-tuned the hyperparameters using the Hyperband algorithm [33]. The final selected model was named DeepEcho3D.

Our findings highlight DeepEcho3D’s ability to improve surface echo detection, reducing outliers by up to 98% compared to traditional threshold methods, which are prone to noise-induced errors. Robust detection of surface echoes is essential for TOF-based surface reconstruction methods used in adaptive imaging for NDT. The developed CNN model addresses this need. Moreover, the model inference time for a single volume is approximately 18 ms, demonstrating its suitability for practical applications.

This work also underscores the importance of automating data annotation, a process that not only improves training efficiency but also reduces human-induced biases, ensuring greater reproducibility.

Looking forward, while DeepEcho3D has shown good results with simple surface geometries, future research should explore its application to more complex morphologies. This will help assess whether training on simple geometries imposes limitations or if the model can generalize effectively to more challenging scenarios. Furthermore, DeepEcho3D was trained to work with a specific probe, a 3 MHz 11 × 11 matrix array. Extending its capabilities to probes with different frequencies and array geometries represents another subject for future research. Ultimately, this research not only advances the field of ultrasonic testing but also lays a foundation for the broader application of deep learning in complex signal processing tasks, potentially benefiting a wide range of industries.

## Figures and Tables

**Figure 1 sensors-25-05033-f001:**
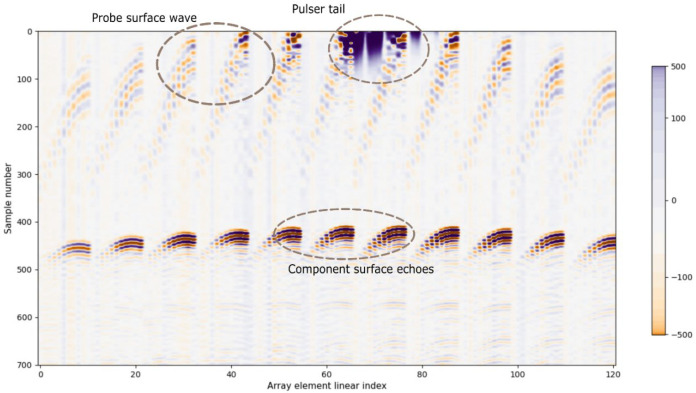
Example of the kind of signals processed in this work, highlighting the echoes that might arrive earlier than the component surface echoes. This B-scan image was acquired with an 11 × 11 matrix array, thus the discontinuities in the image, which correspond to change of elements row.

**Figure 2 sensors-25-05033-f002:**
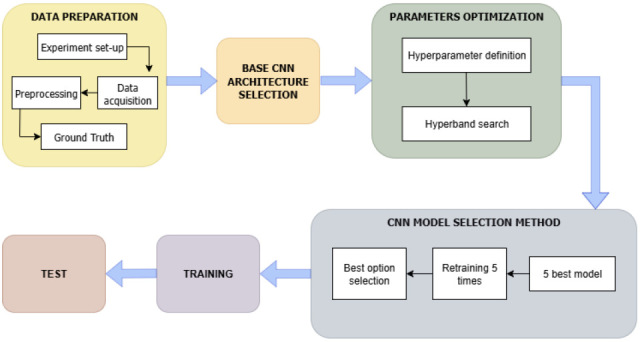
Methodology overview of this study.

**Figure 3 sensors-25-05033-f003:**
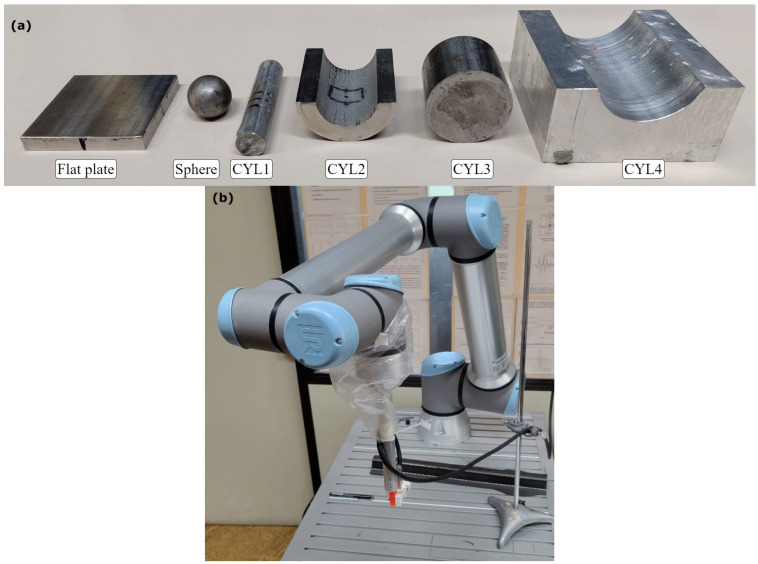
(**a**) Components used in the study. (**b**) Setup robot.

**Figure 4 sensors-25-05033-f004:**
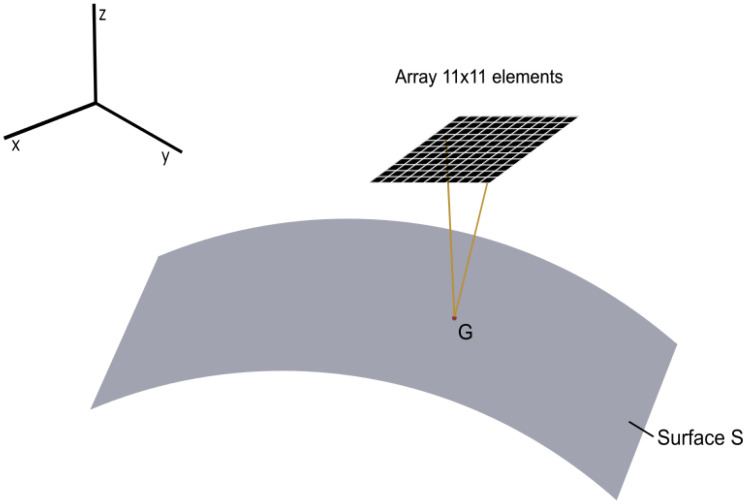
Schematic of the problem, an array probe, and the surface of the component under test.

**Figure 5 sensors-25-05033-f005:**
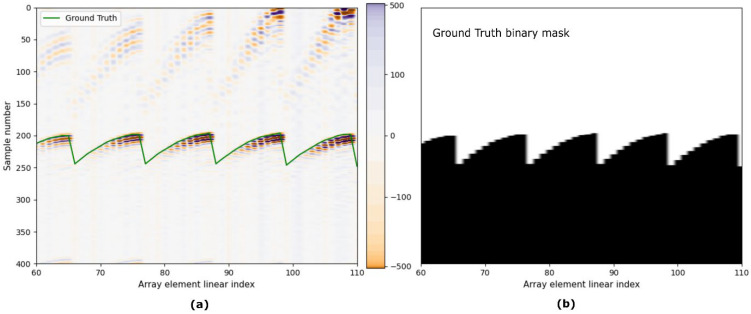
Image example (**a**) and GT binary mask (**b**) generated with the theoretical TOF.

**Figure 6 sensors-25-05033-f006:**
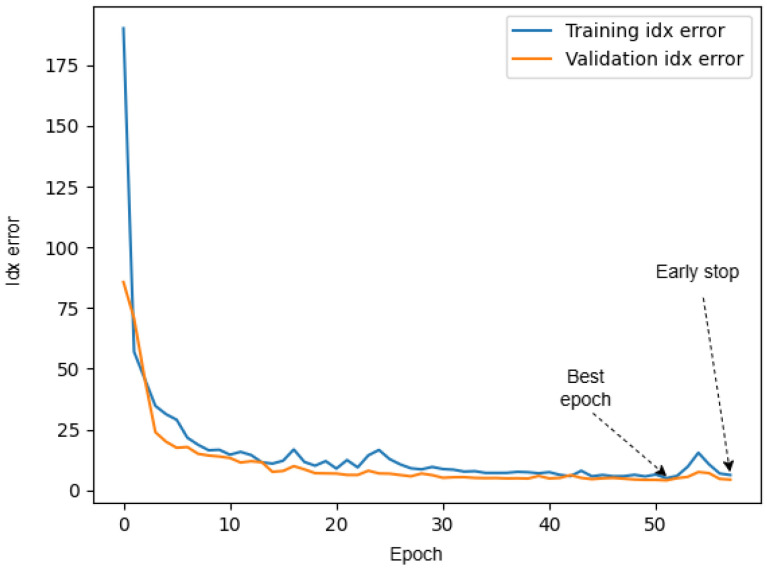
Learning curves monitoring *Idx Error*.

**Figure 7 sensors-25-05033-f007:**
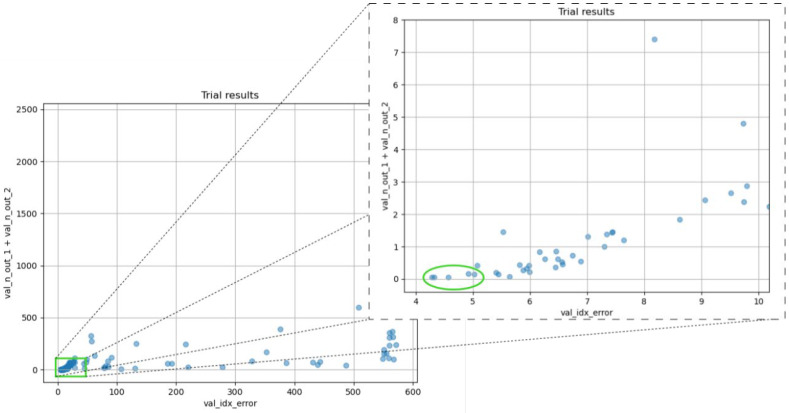
Representation of the hyperparameter search results. Each dot represents a trial.

**Figure 8 sensors-25-05033-f008:**
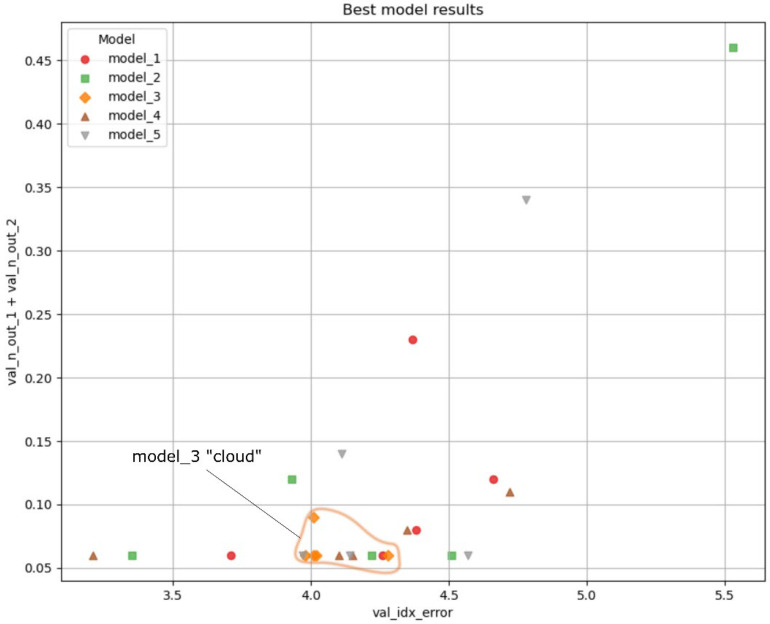
Analysis of 5 top models. The orange curve contains model_3 results. The other four models have a significantly wider spread.

**Figure 9 sensors-25-05033-f009:**
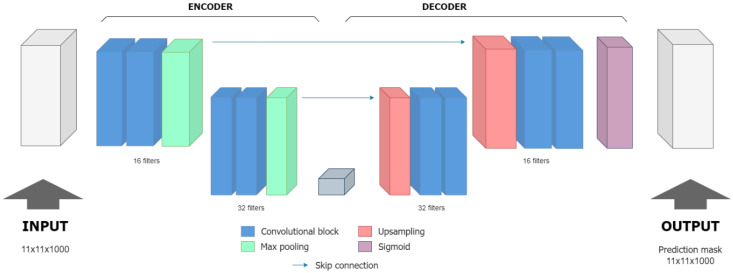
DeepEcho3D architecture. Intermediate activation functions are ReLU.

**Figure 10 sensors-25-05033-f010:**
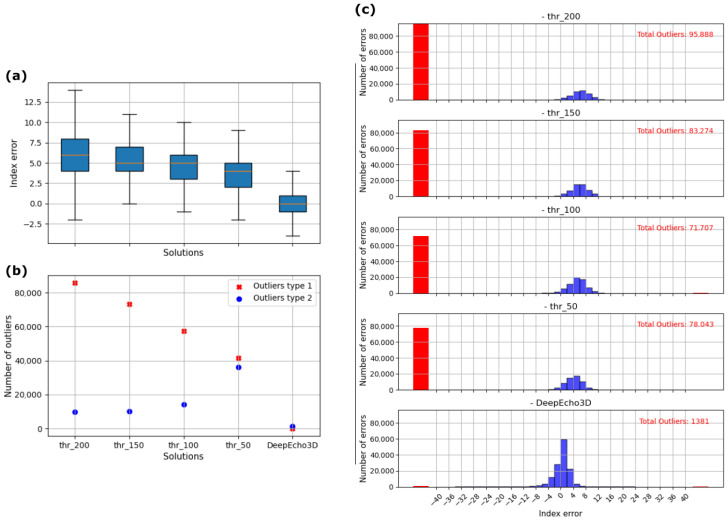
(**a**) Statistics of index error in different solutions; (**b**) Number of outliers; (**c**) Histogram comparing index errors.

**Figure 11 sensors-25-05033-f011:**
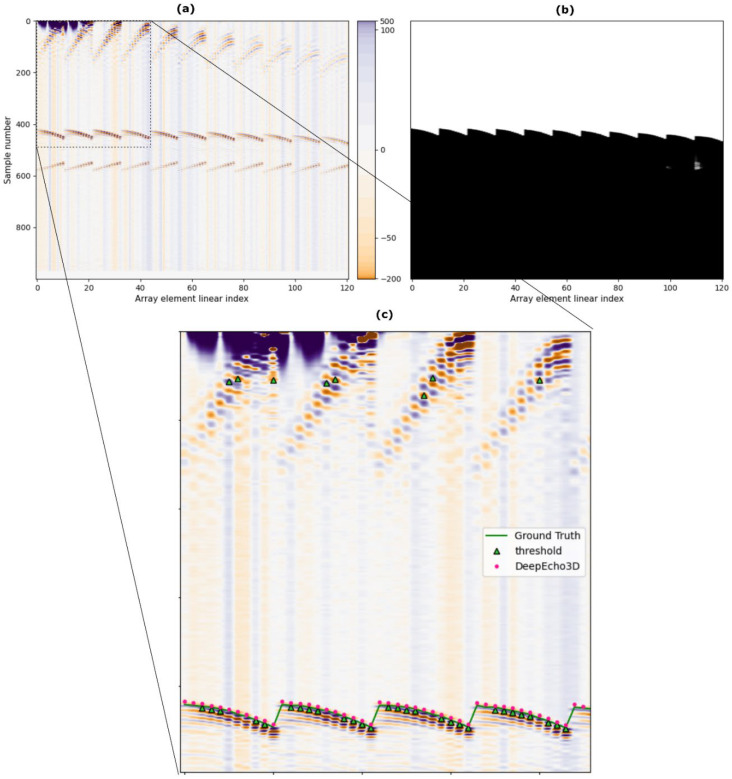
Example of DeepEcho3D response in CYL3. (**a**) Input image, (**b**) CNN output, (**c**) Detail with the threshold detection and DeepEcho3D detection.

**Figure 12 sensors-25-05033-f012:**
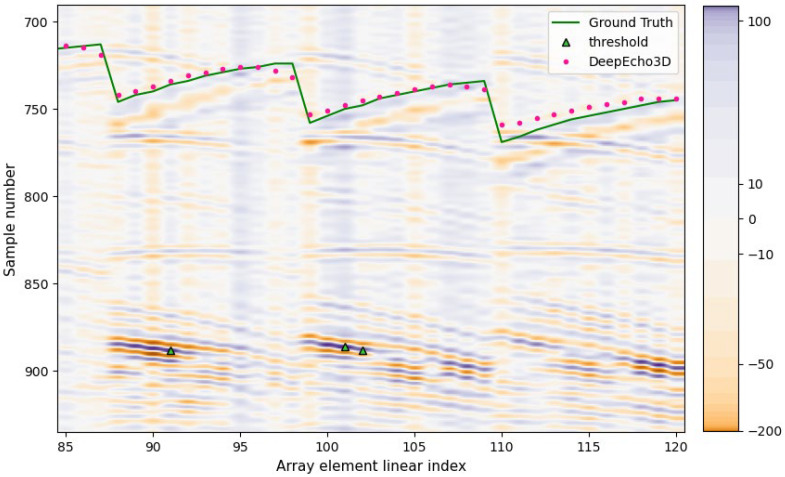
DeepEcho3D performance in CYL4 with very low-amplitude surface echoes, B-Scan vertical range is limited to the region of interest for clarity.

**Figure 13 sensors-25-05033-f013:**
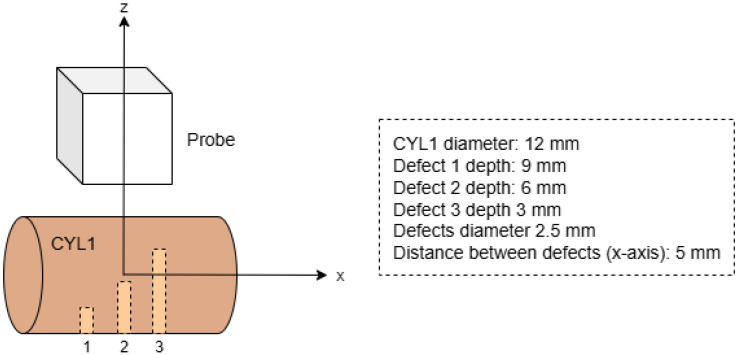
CYL1 schematic showing the three FBH and probe positions. The defect depths shown are measured from the surface of the cylinder.

**Figure 14 sensors-25-05033-f014:**
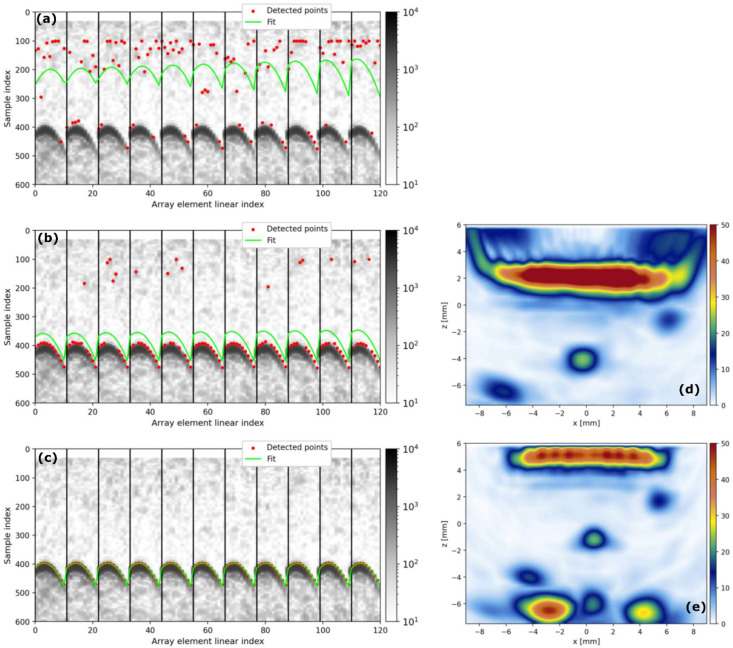
Effect of the surface echo TOF measurement method on a TFM image of the CYL1 test component: (**a**) using 3 × RMS threshold, (**b**) using 6 × RMS threshold, and (**c**) detected by DeepEcho3D. Figure (**d**) presents the D-scan view of the corresponding TFM volumetric image using 6 × RMS surface detection and (**e**) using DeepEcho3D.

**Figure 15 sensors-25-05033-f015:**
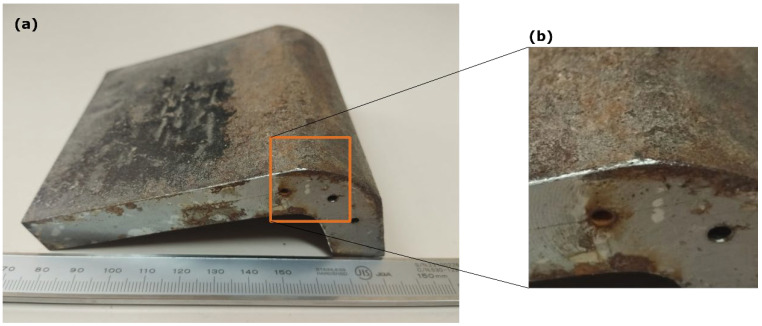
(**a**) Iron component with three side-drilled holes used to test DeepEcho3D generalization capabilities, (**b**) Details of the imaged region. The presence of oxide and surface roughness can be observed.

**Figure 16 sensors-25-05033-f016:**
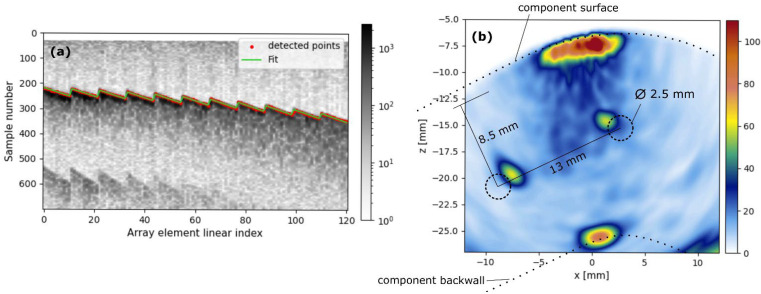
Results from the iron component. (**a**) Pulse-echo B-scan image where the surface echoes can be observed along DeepEcho3D detection results, (**b**) D-scan view extracted from the reconstructed TFM volumetric image.

**Table 1 sensors-25-05033-t001:** Table of components used in the study.

	Diameter	Use	Curvature Type
Flat Plate	-	Train–Validation	Flat
Sphere	19 mm	Train–Validation	Convex
CYL1	12 mm	Test	Convex
CYL2	25 mm	Train–Validation	Concave
CYL3	35 mm	Test	Convex
CYL4	40 mm	Test	Concave

**Table 2 sensors-25-05033-t002:** Table of the five best models’ hyperparameters.

Candidate	Model_1	Model_2	Model_3	Model_4	Model_5
**Pool size**	[1, 1, 8]	[2, 2, 8]	[1, 1, 8]	[2, 2, 8]	[1, 1, 8]
**Number of convolutional blocks**	2	2	2	2	3
**Kernel size**	[3, 3, 18]	[3, 3, 18]	[3, 3, 12]	[3, 3, 18]	[3, 3, 6]
**Number of filters**	16	16	16	16	16
**Learning rate**	0.001	0.001	0.001	0.001	0.001
**Loss function**	Binary cross-entropy	Binary cross-entropy	Binary cross-entropy	Tversky	Binary cross-entropy

**Table 3 sensors-25-05033-t003:** Table of stability results.

Candidate	Centroid Idx Error	Centroid Outliers	Avg. Distance
Model_1	4.28	0.11	0.25
Model_2	4.31	0.15	0.59
Model_3	4.06	0.06	0.09
Model_4	4.11	0.07	0.36
Model_5	4.31	0.13	0.30

**Table 4 sensors-25-05033-t004:** Table of segmentation test metrics.

Metric	Mean
Dice	0.993
IOU	0.988
Recall	0.997
Precision	0.991
Accuracy	0.996

**Table 5 sensors-25-05033-t005:** Table of solution test metrics.

Solution	Error Std	95% Error Interval	Outliers (%)
Thr 200	327	[−849, 10]	70
Thr 150	332	[−847, 10]	61
Thr 100	326	[−840, 9]	52
Thr 50	319	[−829, 8]	57
DeepEcho3D	16	[−9, 5]	1

## Data Availability

The data presented in this study are available on request from the corresponding author.

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
