# Peer review of "Three-Dimensional Convolutional Neural Network for Ultrasound Surface Echo Detection"

_sensors, 2025, doi:10.3390/s25165033_

Round 1
Reviewer 1 Report
Comments and Suggestions for Authors
In Section 3.1, when evaluating the strengths and weaknesses of the model, it is necessary to consider not only the validation error and outliers, but also parameters such as computational accuracy and training time to comprehensively assess the model's performance. Please elaborate on this.Which five model architectures are specifically referred to as the best models introduced in Section 3.2? Please provide detailed explanations. Furthermore, a scatter plot does not allow for a clear visual assessment of the performance levels of the five models. What criteria are used to evaluate the strengths and weaknesses of these models? Is it based on the sum of the values represented by the two coordinates, or do they each have their own weights? How is this judgment made without proper comparative standards? It is recommended to provide additional clarification here and to offer a comprehensive evaluation of the model's performance.
The parameter settings in Table 2 have already been described in detail in the text; is there still a need for their presence? Alternatively, could the parameter settings for the convolutional network be presented directly in a table without redundant narration? Moreover, is the term "DeepEcho3D" derived from the parameter settings described above? Does it imply that merely altering the foundational architecture of the convolutional network can lead to a new terminology?
Why was the Sigmoid function chosen as the activation function in the CNN architecture shown in Figure 9? Why was a simpler computational function not selected instead?
In Section 3.4, different threshold scenarios are compared with the selected models. What is the basis for choosing specific thresholds? Is there a relationship between the chosen thresholds and the selected models? Randomly selected thresholds can lead to inconsistent model performance, making it difficult to effectively assess the true capabilities of the model, and such thresholds may not accurately reflect the true distribution of the data.
Section 3.5 presents the TFM example of "CYL1"; however, earlier sections mentioned six different experimental subjects with varying surface morphologies, and the text does not provide a sufficient comparison of these different morphological objects. How can the model's performance on objects with different surface morphologies be validated? It is recommended to include additional experiments to thoroughly validate the model's performance.
Author Response
We thank the reviewer for these insightful comments, which have helped us significantly improve the clarity and rigor of our manuscript. We have addressed all the points raised, as detailed below.
In Section 3.1, when evaluating the strengths and weaknesses of the model, it is necessary to consider not only the validation error and outliers, but also parameters such as computational accuracy and training time to comprehensively assess the model's performance. Please elaborate on this.
We agree that computational cost is an important factor. We have now included a discussion on the training time in Section 3.3. The Hyperband algorithm inherently addresses this by efficiently discarding underperforming configurations, saving significant time compared to an exhaustive search. The final selected model, DeepEcho3D, takes approximately 3 hours to train, and inference on a single volume is performed in approximately 18 milliseconds, making it suitable for practical applications.
Which five model architectures are specifically referred to as the best models introduced in Section 3.2? Please provide detailed explanations.
To clarify the differences between the five best-performing models evaluated in Section 3.2, we have added a new table (Table 2) that details the specific hyperparameter configurations for each of the five models. This table clearly shows the differences in parameters.
Furthermore, a scatter plot does not allow for a clear visual assessment of the performance levels of the five models. What criteria are used to evaluate the strengths and weaknesses of these models? Is it based on the sum of the values represented by the two coordinates, or do they each have their own weights? How is this judgment made without proper comparative standards? It is recommended to provide additional clarification here and to offer a comprehensive evaluation of the model's performance.
We acknowledge that our initial rationale for selecting Model_3 could be more explicit and quantitatively supported. To address the reviewer's valid point regarding the lack of clear comparative standards, we have performed a new quantitative analysis and revised the manuscript accordingly.
We now evaluate the top five models based on their performance stability, presented in a new table (Table 3). Specifically, we calculated the average dispersion for each model's performance cluster across five retraining runs.
The results in this table provide a clear justification: Model_3 exhibits the lowest average dispersion, quantitatively proving it is the most stable and reliable model. We have updated the text in Section 3.3 to present this analysis, which now provides the comparison that was requested.
The parameter settings in Table 2 have already been described in detail in the text; is there still a need for their presence? Alternatively, could the parameter settings for the convolutional network be presented directly in a table without redundant narration? Moreover, is the term "DeepEcho3D" derived from the parameter settings described above? Does it imply that merely altering the foundational architecture of the convolutional network can lead to a new terminology?
We have streamlined the text by removing the redundant narrative description of the final model's parameters and now use Table 2 as the primary summary. We have also clarified in Section 3.3 that the name "DeepEcho3D" refers to the final, optimized model resulting from our complete methodology, not just a new network class. The name is used solely for convenient reference and does not imply the introduction of a new architecture.
Why was the Sigmoid function chosen as the activation function in the CNN architecture shown in Figure 9? Why was a simpler computational function not selected instead?
The choice of the Sigmoid function as the final activation layer is standard practice for binary segmentation tasks, which is how we have framed our problem. The Sigmoid function maps the network's output to a range between [0, 1] for each voxel, which can be interpreted as the probability of that voxel belonging to the "surface and below" class. This is essential for generating the binary mask shown in Figure 5b. The activations on the intermediante layers are ReLU functions, which are computationally simpler. This was clarified in Figure 9 caption.
In Section 3.4, different threshold scenarios are compared with the selected models. What is the basis for choosing specific thresholds? Is there a relationship between the chosen thresholds and the selected models? Randomly selected thresholds can lead to inconsistent model performance, making it difficult to effectively assess the true capabilities of the model, and such thresholds may not accurately reflect the true distribution of the data.
The thresholds used for comparison were not chosen randomly. They were selected to represent a range of practical choices an operator might make, based on the signal's noise background level. The values (50, 100, 150, 200) represent fixed amplitude levels spanning from a low threshold (more sensitive, but prone to noise, as seen in our results) to a high threshold (more robust against noise, but prone to missing weaker echoes). This allows for a comprehensive comparison, demonstrating that DeepEcho3D outperforms the traditional method across a spectrum of reasonable, fixed-level parameter settings. We have added a sentence to clarify this rationale in Section 3.4.
Section 3.5 presents the TFM example of "CYL1"; however, earlier sections mentioned six different experimental subjects with varying surface morphologies, and the text does not provide a sufficient comparison of these different morphological objects. How can the model's performance on objects with different surface morphologies be validated? It is recommended to include additional experiments to thoroughly validate the model's performance.
We agree that demonstrating performance on varied morphologies is important. This concern is partially addressed in Section 3.5, where DeepEcho3D's performance is evaluated on an iron component with surface oxidation and a complex geometry (combined planar and cylindrical regions) that was not seen during training (Figure 15). The successful TFM reconstruction of this component (Figure 16) serves as a practical validation example of the model's ability to generalize to different surface morphologies and conditions beyond the simple, clean geometries used for training. Validation on a broader set of components with greater shape variability will be addressed in future research.
Reviewer 2 Report
Comments and Suggestions for Authors
The work could be interesting. More relevant work should be reviewed and discussed e.g. Miorelli, R., Robert, S., Calmon, P., & Le Berre, S. (2024). Use of deep learning and data augmentation by physics-based modelling for crack characterisation from multimodal ultrasonic TFM images. Nondestructive Testing and Evaluation, 1–23. https://doi.org/10.1080/10589759.2024.2428374;
Tu, J., Wang, H., Song, Y., Wu, Q., Zhang, X., & Song, X. (2025). The characterisation of surface-breaking crack using ultrasonic total focusing method imaging based on COA-VMD. Nondestructive Testing and Evaluation, 1–23. https://doi.org/10.1080/10589759.2025.2489597.
More dataset preparation and robustness should be discussed.
Author Response
The work could be interesting. More relevant work should be reviewed and discussed e.g. Miorelli, R., Robert, S., Calmon, P., & Le Berre, S. (2024). Use of deep learning and data augmentation by physics-based modelling for crack characterisation from multimodal ultrasonic TFM images. Nondestructive Testing and Evaluation, 1–23. https://doi.org/10.1080/10589759.2024.2428374;
Tu, J., Wang, H., Song, Y., Wu, Q., Zhang, X., & Song, X. (2025). The characterisation of surface-breaking crack using ultrasonic total focusing method imaging based on COA-VMD. Nondestructive Testing and Evaluation, 1–23. https://doi.org/10.1080/10589759.2025.2489597.
We thank the reviewer for pointing out these relevant and recent works. We have read the suggested papers by Miorelli et al. (2024) and Tu et al. (2025) and have now included a discussion of them in our literature review (Introduction). These works focus on crack characterization using TFM images and advanced signal processing, while our work addresses the prior, fundamental step of robust surface detection needed to generate accurate TFM images in the first place, especially in auto-focusing contexts. The revised introduction now better situates our contribution within the very latest developments in the field.
More dataset preparation and robustness should be discussed.
We have added a quantitative stability analysis (Table 4) in Section 3.3 to prove the model's robustness against the randomness of the training process. This analysis confirms that the selected architecture is consistently the most stable.
Additionally, our test on a component with surface oxidation (Figure 16), which was not part of the training data, serves as a practical demonstration of the model's generalization capabilities to more realistic conditions. We believe these elements now provide a more thorough demonstration of the model's robustness.
Reviewer 3 Report
Comments and Suggestions for Authors
This paper is suitable for publication as it reports the application results of reference [16]. However, in order to clearly demonstrate the superiority of the authors' proposed method, "DeepEcho3D," the following points should be addressed:
1. On page 4, you describe the test components used. Please explain why these specific components are appropriate for evaluating your method. If this has already been discussed in a previous publication, please provide a citation to that paper.
2. According to Figure 8, you selected "Model 3" among the top five models. Please describe the differences among these five models to clarify the rationale for your selection.
3. The ultrasonic transducer used operates at 3 MHz, and the sampling frequency is 40 MHz. Please comment on whether this sampling frequency is appropriate for the transducer frequency and provide a brief justification.
Author Response
We thank the reviewer for their positive feedback and for the constructive comments, which have helped us improve the clarity and justification of our methodology. We have addressed each of the points raised.
This paper is suitable for publication as it reports the application results of reference [16]. However, in order to clearly demonstrate the superiority of the authors' proposed method, "DeepEcho3D," the following points should be addressed:
- On page 4, you describe the test components used. Please explain why these specific components are appropriate for evaluating your method. If this has already been discussed in a previous publication, please provide a citation to that paper.
Thank you for this question. The components were deliberately chosen to represent fundamental, canonical surface shapes encountered in NDT: flat, convex, and concave. This allowed for a controlled and systematic evaluation of the model's ability to learn these basic geometric features before testing on more complex, real-world components (like the oxidized part in Figure 15). We have added a sentence to Section 2.1.1 to clarify this.
- According to Figure 8, you selected "Model 3" among the top five models. Please describe the differences among these five models to clarify the rationale for your selection.
We thank the reviewer for this important question, which was also raised by another reviewer. We agree that a clearer, data-driven rationale for our model selection was needed. To address this, we have made two key additions to the manuscript:
-Describing Model Differences: To clarify the architectural differences, we have modified Table 2 to now detail the specific hyperparameter configurations of all top five candidate models.
-Clarifying Selection Rationale: To provide a comparative standard, we have added a new quantitative stability analysis to the manuscript. This analysis, presented in a new Table 3, evaluates each model's stability using its performance dispersion across five retraining runs. The revised text in Section 3.3 now uses the results from this new table to quantitatively demonstrate that Model_3 is the most stable candidate.
We hope these additions provide the clarity and data-driven justification that the reviewer rightly requested.
- The ultrasonic transducer used operates at 3 MHz, and the sampling frequency is 40 MHz. Please comment on whether this sampling frequency is appropriate for the transducer frequency and provide a brief justification.
This is indeed an important point. A common rule of thumb for selecting an appropriate sampling frequency is to use at least four times the transducer's center frequency. This helps avoid aliasing in the upper end of the transducer’s bandwidth, which in this case extends up to approximately 6 MHz (assuming a 100% bandwidth centered at 3 MHz). Therefore, the minimum theoretical sampling rate should be around 12 MHz. The 40 MHz sampling frequency used in our setup is well above this requirement, ensuring accurate signal acquisition. While using a lower sampling rate could reduce memory usage and computational cost, exploring such optimizations is left for future work.
Round 2
Reviewer 2 Report
Comments and Suggestions for Authors
This manuscript develops a robust method based on a 3D CNN (DeepEcho3D) for surface echo detection in ultrasound Full Matrix Capture (FMC) data acquired with 2D matrix arrays. It addresses the limitations of traditional threshold-based methods (noise sensitivity and high outlier rates) and improves Time-of-Flight (TOF) measurement accuracy. The results are well-supported and constitute a clear contribution to the field of ultrasonic NDT. The reviewer recommends Minor Revision. Specific issues:
- Since a 2D matrix array was used, why not present 3D imaging results?
- Was signal drift correction considered during raw data processing? The reviewer notes apparent baseline variations in Figure 1.
- Figure 15 shows three defects, yet the imaging results (Fig 16b) appear to detect only two.
- Please mark the actual locations and sizes of the defects in Figure 16.
- Provide quantitative error analysis between the ultrasonic imaging results and the actual defect positions.
- According to Section 2.1.2, each FMC acquisition yields 121 images of 121*1000 pixels, reshaped into an 11*11*1000 input volume. What was the total number of training volumes used?
- Pleaseprovide a quantitative discussion of the TOF prediction error (difference between predicted and actual arrival times).
- What is the prediction efficiency? Does it meet real-time inspection requirements?
- Pleaseredraw Figure 13 to include clear dimensional and positional annotations.
English proofreading is expected
Author Response
This manuscript develops a robust method based on a 3D CNN (DeepEcho3D) for surface echo detection in ultrasound Full Matrix Capture (FMC) data acquired with 2D matrix arrays. It addresses the limitations of traditional threshold-based methods (noise sensitivity and high outlier rates) and improves Time-of-Flight (TOF) measurement accuracy. The results are well-supported and constitute a clear contribution to the field of ultrasonic NDT. The reviewer recommends Minor Revision. Specific issues:
- Since a 2D matrix array was used, why not present 3D imaging results?
We thank the reviewer for this pointing this out. The imaging results presented are indeed 2D slices (D-scans) extracted from the full 3D TFM volumetric image. This is a common practice for clarity of visualization in a manuscript, as rendering a full 3D volume can be confusing due to occlusions. We have added a sentence to the figure caption to make this explicit. -
Was signal drift correction considered during raw data processing? The reviewer notes apparent baseline variations in Figure 1.
We thank the reviewer for this observation. The reviewer is correct in noting the presence of minor baseline variations in the raw signal data presented in Figure 1.This was addressed during pre-processing. As stated in Section 2.1.2, we applied a band pass filter with [0.5, 6] MHz band. The high-pass component of this filter at 0.5 MHz is designed to effectively remove such low-frequency baseline drift.
Furthermore, it is worth highlighting that even if minor baseline variations persist after filtering, one of the key strengths of our data-driven approach is the CNN's ability to learn and discriminate against such artifacts. The model learns to identify the specific waveform of the surface echo while ignoring irrelevant low-frequency variations. The high accuracy and low outlier rate demonstrated in our results (Section 3.4) confirm that DeepEcho3D successfully handles these real-world signal imperfections.
- Figure 15 shows three defects, yet the imaging results (Fig 16b) appear to detect only two.
The reviewer is correct in observing that only two of the three SDHs are visible in the presented D-scan (Fig 16b). The third hole is outside of the “shadow” of the array aperture, and that’s why it is not visible in the image. - Please mark the actual locations and sizes of the defects in Figure 16.
Thank you for this suggestion. We have revised Figure 16b to include dashed circles indicating the nominal locations and sizes of the SDHs, allowing for a direct visual comparison with the imaging results. - Provide quantitative error analysis between the ultrasonic imaging results and the actual defect positions.
We appreciate the reviewer’s suggestion. The side drilled holes (SDH) indications shown in Figure 16 correspond to portions of the hole edges, as the wavelength is comparable to the hole diameter (2.5 mm). Consequently, the centers of the SDHs cannot be precisely identified in the TFM image. However, the dashed circles added in Figure 16 represent the actual SDH locations, and the observed positional discrepancies are on the order of approximately 1 mm. - According to Section 2.1.2, each FMC acquisition yields 121 images of 121*1000 pixels, reshaped into an 11*11*1000 input volume. What was the total number of training volumes used?
As stated in Section 2.5, the total dataset consists of 6000 volumes, split 70:15:15 for training, validation, and testing. Therefore, the total number of volumes used for training was 4200 (70% of 6000). - Please provide a quantitative discussion of the TOF prediction error (difference between predicted and actual arrival times).
We provide a detailed quantitative analysis of the TOF prediction error in Section 3.4. Specifically, Figure 10 shows the box plots and histograms of the index error distribution, and Table 4 provides key statistical metrics, including the standard deviation of the error (16 samples) and the 95% error interval ([-9, 5] samples) for DeepEcho3D. These results quantitatively demonstrate the low prediction error of our method. - What is the prediction efficiency? Does it meet real-time inspection requirements?
The prediction efficiency is detailed at the end of Section 3.3. The inference time for a single FMC volume is approximately 18 ms on an NVIDIA 2080 Ti GPU. While the definition of 'real-time' depends on the acquisition speed of the specific inspection system, this high processing speed is suitable for many NDT applications, particularly for offline data analysis or for online systems where the acquisition rate is not excessively high. A sentence regarding inference time has been included in the Conclusions section. - Please redraw Figure 13 to include clear dimensional and positional annotations.
Thank you for the suggestion. We have redrawn Figure 13 to include clear dimensional and positional annotations for the three Flat Bottom Holes (FBHs), as requested.